# Patterns of Red and Processed Meat Consumption across North America: A Nationally Representative Cross-Sectional Comparison of Dietary Recalls from Canada, Mexico, and the United States

**DOI:** 10.3390/ijerph18010357

**Published:** 2021-01-05

**Authors:** Sarah M. Frank, Lindsay M. Jaacks, Carolina Batis, Lana Vanderlee, Lindsey Smith Taillie

**Affiliations:** 1Carolina Population Center, University of North Carolina at Chapel Hill, Chapel Hill, NC 27516, USA; sfrank@unc.edu; 2Department of Nutrition, Gillings School of Global Public Health, University of North Carolina at Chapel Hill, Chapel Hill, NC 27599, USA; 3Global Academy of Agriculture and Food Security, Easter Bush Campus, The University of Edinburgh, Roslin EH25 9RG, UK; lindsay.jaacks@ed.ac.uk; 4CONACYT—Health and Nutrition Research Center, National Institute of Public Health, Av. Universidad No. 655 Colonia Santa María Ahuacatitlán, Cuernavaca 62100, Mexico; carolina.batis@insp.mx; 5Centre Nutrition, Santé et Société (NUTRISS), L’École de Nutrition, Université Laval, Quebec City, QC GIV 0A6, Canada; lana.vanderlee@fsaa.ulaval.ca

**Keywords:** consumer behavior, nutrition policy, meat, diet surveys, environment and public health, cross-sectional study, socioeconomic factors, Canada, Mexico, United States

## Abstract

Close economic ties encourage production and trade of meat between Canada, Mexico, and the US. Understanding the patterns of red and processed meat consumption in North America may inform policies designed to reduce meat consumption and bolster environmental and public health efforts across the continent. We used nationally-representative cross-sectional survey data to analyze consumption of unprocessed red meat; processed meat; and total red and processed meat. Generalized linear models were used to separately estimate probability of consumption and adjusted mean intake. Prevalence of total meat consumers was higher in the US (73.6, 95% CI: 72.3–74.8%) than in Canada (65.6, 63.9–67.2%) or Mexico (62.7, 58.1–67.2%). Men were more likely to consume unprocessed red, processed, and total meat, and had larger estimated intakes. In Mexico, high wealth individuals were more likely to consume all three categories of meat. In the US and Canada, those with high education were less likely to consume total and processed meat. Estimated mean intake of unprocessed red, processed, and total meat did not differ across sociodemographic strata. Overall consumption of red and processed meat remains high in North America. Policies to reduce meat consumption are appropriate for all three countries.

## 1. Introduction

High levels of both unprocessed red and processed meat have been associated with elevated risk for colorectal, stomach, and pancreatic cancers, with the strongest evidence for processed meat [1]. High intake of these foods is also associated with obesity [2], diabetes [3], and heart disease [3]. In addition to health concerns, red meat production is a major source of greenhouse gas emissions, deforestation, and water usage [4]. In light of both the health and environmental impacts of red and processed meat consumption, recent public health recommendations have called for individuals to reduce or eliminate intake of these foods [5,6].

Policies and trade agreements such as the North American Free Trade Agreement (NAFTA) and its successor, the US–Mexico–Canada Agreement (USMCA), encourage high production, trade, and ultimately consumption of many food products, including red and processed meat [7,8]. Indeed, market data indicate that purchases of red and processed meat are above the global average of 29 g/capita/day in all three North American countries [9].

On the other hand, policy strategies can be used to discourage consumption of food groups of concern. For example, taxes have been successfully used to reduce purchases of sugar-sweetened beverages in over 35 countries, including Mexico, as well as in several US cities [10]. Responses to public policy interventions such as food and beverage taxes differ by socioeconomic status and importantly, such policies are most effective among the highest consumers [11,12]. While information on dietary intake of meat is available in the US [13] and Mexico [14] or as a contributor to nutrient intake in Canada [15], no studies have specifically examined patterns of red and processed meat consumption across these North American countries using a common methodology. Previous multi-country comparative studies are limited by their use of commodity data from Food Balance Sheets [16], but such data do not identify top consumers, nor correlate well with actual consumption [16]. Understanding the patterns of red and processed meat intake at the individual level can be used to identify the highest consumers and inform policy. The objectives of this study were to use nationally-representative 24-h dietary recall data from Canada, Mexico, and the US to compare and contrast sociodemographic correlates of red and processed meat consumption throughout North America. It was hypothesized that meat intake would be higher in Canada and the US than in Mexico, and that it would be higher for men than for women in all three countries. Additionally, it was hypothesized that education would be a stronger correlate of meat intake in Canada and the US, while wealth would be more strongly correlated with meat intake in Mexico.

## 2. Materials and Methods

### 2.1. Data Sources

#### 2.1.1. Canada: Canadian Community Health Survey 2015

The Canadian Community Health Survey (CCHS)-Nutrition focus was most recently administered in 2015 [17]. CCHS-Nutrition used a three-stage sampling design to establish a representative sample of the civilian, non-institutionalized population in all provinces, except Prince Edward Island. The sample excluded residents of the Territories, reserves, and other Aboriginal settlements. To account for seasonality, selected clusters in each province were randomized to one of six two-month data collection periods. Face-to-face interviews were administered in participants’ homes. Cross-sectional data were from 14,083 participants aged 18 years or older with one day of valid dietary intake data. More detailed information on recruitment, sample design, and accounting for missing or invalid data is available in Appendix A.

#### 2.1.2. Mexico: National Health and Nutrition Survey 2016

The Mexican National Health and Nutrition Survey (Spanish acronym, ENSANUT) was administered by the National Institute for Public Health (INSP) from May to October 2016 [18]. ENSANUT used a multi-stage probability design to sample the civilian, non-institutionalized population of Mexico. Face-to-face interviews with demographic, socioeconomic, dietary, and health-related questions were conducted in participants’ homes. Cross-sectional data were from 1581 participants aged 18 years or older with one day of valid dietary intake data. More detailed information on recruitment, sample design, and accounting for missing or invalid data is available in Appendix A.

#### 2.1.3. US: National Health and Nutrition Examination Survey 2013–2016

The US National Health and Nutrition Examination Survey (NHANES) is a repeated cross-sectional survey that uses multistage probability design to sample the civilian, non-institutionalized population residing in the 50 states and District of Columbia [19]. Data were collected in the southern US from 1 November to 30 April and in the northern regions from 1 May to 31 October. Face-to-face demographic and socioeconomic interviews were conducted in participants’ homes. Face-to-face dietary and health-related questionnaires were administered in a private room in the Mobile Examination Center. Data were from 10,497 participants aged 18 years or older who participated in either the 2013–2014 or the 2015–2016 NHANES cycle and who had one day of valid dietary intake data according to the National Center of Health Statistics. More detailed information on recruitment, sample design, and accounting for missing or invalid data is available in Appendix A.

### 2.2. Dietary Recalls and Intake of Red and Processed Meat

For each survey, trained interviewers used the US Department of Agriculture (USDA) Automated Multiple Pass Method to gather 24-h dietary recall data on individual food intake, as described elsewhere [20]. Participants were asked to recall all foods and beverages they consumed the previous day. Measuring guides were used to assist with approximating the portion sizes of consumed foods. Thus, the reported dietary intake across countries is directly comparable. For CCHS and ENSANUT, the Automated Multiple Pass Method was applied accounting for the country-specific context (e.g., the list of commonly forgotten foods was adapted and differences in food availability and preparations were considered) [17,18].

The 24-h dietary recall data from the US and Canada were merged with the USDA Food Patterns Equivalent Database, which disaggregates mixed dishes into their component ingredients’ gram weights using standard recipes [21,22]. For Mexico, a standard recipe file created by INSP was applied to mixed dishes. In addition to gram weights of meat intake, the recipe files were also used to calculate single-day energy intake by summing kilocalories for every food and beverage item reported.

The same, standardized definition was applied across countries to identify unprocessed red meat (defined as mammalian muscle meat, e.g., beef, veal, pork, lamb, mutton, and goat) and processed meat (defined as all meats that have been salted, cured, fermented, smoked, or otherwise processed for preservation and flavor enhancement). The probability of consumption (i.e., proportion of consumers) and the amount consumed (g/day) among consumers of (1) unprocessed red meat; (2) processed meat; and (3) total red and processed meat was estimated. Amount consumed (g/day) was estimated using cooked weight.

### 2.3. Sociodemographic Correlates

#### 2.3.1. Canada, CCHS

Self-reported household income and number of household members were used to divide individuals into low, middle, and high income tertiles of per-capita income. Education was classified as low (high school equivalent or lower), middle (trade or college certificate), and high (university degree or higher).

#### 2.3.2. Mexico, ENSANUT

Due to insufficient data on household income, and because income is not considered a reliable measure of financial assets in middle-income countries [23], a wealth index was used in Mexico. Principal component analysis was used to create a wealth index using information on household dwelling characteristics, basic services, and material goods. This index was used to classify participants into low, medium, and high wealth categories. Household size was defined as the total number of persons that lived in the dwelling and was included as a variable in regression analyses since a measure of per-capita income was not available. Education was classified as low (primary school or less), middle (more than primary but less than high school), and high (high school or higher).

#### 2.3.3. US, NHANES

The Poverty Income Ratio (PIR), a measure of family income relative to the Federal Poverty Level that takes into account household size, was used to create wealth categories. Family income was categorized as low (PIR 0–130%), middle (PIR 131–399%), and high (PIR ≥ 400%). Education was classified as low (high school equivalent or lower), middle (some college), and high (college degree or higher).

### 2.4. Statistical Analyses

All analyses accounted for complex survey design and were performed using Stata 15 (StataCorp LLC, College Station, TX, USA). Descriptive statistics were used to determine the percentage of consumers and median consumption among meat-consumers. Rao-Scott chi-square tests were used to compare the proportion of meat consumers between countries. Somer’s D was used to test for differences in unadjusted median intake of unprocessed red, processed, and total meat between countries.

In analyses of sociodemographic characteristics, multivariate negative binomial regression was used to estimate the adjusted predicted probability of consuming red, processed, and total meat by sex, age (10-year categories), educational attainment, and household income and wealth. For analyses of ENSANUT only, we further adjusted for household size in order to account for sharing of wealth across household members. Stata’s postestimation margins command, dydx option was used to compare likelihood of meat consumption by sociodemographic characteristics. Generalized linear models were used to estimate the multivariate adjusted mean consumption of unprocessed red, processed, and total meat among consumers only, overall and within levels of the aforementioned covariates. In sensitivity analyses, we further adjusted for total energy intake in kilocalories. All sociodemographic analyses were stratified by country.

## 3. Results

### 3.1. Sample Characteristics

The final sample included 26,161 non-pregnant adults across the three North American countries (Appendix A). The overall percentage (95% CI) of single-day total unprocessed and processed meat consumers was lower in Canada (65.6 (63.9–67.2%)) and Mexico (62.7 (58.1–67.1%)) than it was in the US (73.6 (72.3–74.8%), *p* < 0.001, Table 1). The proportion of processed meat consumers was also lower in Canada (36.3 (34.6–38.0%)) and Mexico (30.6 (25.9–35.8%)) than it was in the US (47.1 (45.7–48.5%), *p* < 0.001). The overall prevalence of single day unprocessed red meat consumption was similar across the three countries.

For total unprocessed red and processed meat, the median (IQR) grams of total meat among meat-consumers was higher in Canada (79.0 (36.2–140.1) grams) than in Mexico (62.5 (31.3–117.4) grams, *p* = 0.04), but not different from median intake in the US (79.4 (40.8–134.7) grams). The median (IQR) grams of unprocessed red meat was higher in Canada (79.0 (36.6–131.6)), than in the US (72.3 (38.3–124.5) grams, *p* = 0.04). Median intake of unprocessed red meat was 62.0 (28.7–114.8) grams in Mexico and was not significantly different from the other two countries, though confidence intervals were wide. The amount of processed meat consumed was slightly lower than unprocessed red meat and similar for all three countries: 41.8 (21.2–82.4) grams in Canada, 40.0 (20.0–76.9) grams in Mexico, and 44.5 (17.9–84.2) grams in the US.

Results from sensitivity analyses are available in Appendix A. Results herein are from fully-adjusted models including sex, age category, educational attainment, wealth and income category, and, in Mexico, household size.

### 3.2. Total Meat

In all three countries, the likelihood of consuming total meat was higher for men than for women (13.1 (9.9–16.4), 9.2 (1.4–17.0), and 9.4 (6.7–12.0) percentage points higher in Canada, Mexico, and the US, respectively) (Figure 1). Among consumers, estimated average total meat intake was higher for men than women in Canada and the US but not for Mexico, though the confidence intervals were wide (Figure 2).

The likelihood of consuming total meat was 37.4 (14.8–60.0) percentage points lower for individuals older than 75 in Mexico, relative to 18–24-year-old adults. There was no meaningful difference in predicted probability of meat consumption by age in Canada or the US. However, the estimated average intake of total meat was lower for the oldest adults in Canada and the US (29.6 (12.5–46.7) and 22.7 (11.6–33.8) grams lower, respectively).

For education, in Canada and the US, those with low and middle education were more likely to consume total meat than those with high education (in Canada, 6.6 (2.1–11.0) and 6.6 (2.2–11.0) percentage points higher for low and middle education, respectively, and in the US, 5.6 (2.3–8.9) and 5.0 (1.9–8.1) percentage points higher, respectively). There were no associations between education and likelihood of total meat consumption in Mexico.

There were no associations between income and likelihood of total meat consumption in Canada. Individuals with low and middle wealth in Mexico were less likely to consume total meat, relative to those with high wealth (23.5 (14.0–32.9) and 11.6 (2.0–21.3) percentage points lower for low and middle wealth, respectively). In the US, the predicted probability of consuming total meat was 5.8 (3.3–8.3) percentage points higher for those with middle income, relative to those with high income. Adjusted mean intake of total meat among those who consumed meat did not differ by wealth and income or education within any country.

### 3.3. Unprocessed Red Meat

Men were more likely than women to consume unprocessed red meat in Canada and the US (7.8 (4.2–11.3) and 8.6 (5.4–11.8) percentage points higher, respectively), but not in Mexico (Appendix A). Among consumers, estimated mean unprocessed red meat intake was higher for men than women in Canada and the US, but not for Mexico.

In Canada, the predicted probability of unprocessed red meat consumption was 4.1 (0.1–8.7) percentage points higher for individuals with low education, relative to those with high education. In the US, both low and middle education were associated with a greater likelihood of unprocessed red meat consumption (7.0 (2.5–11.5) and 3.9 (0.2–7.8) percentage points higher for low and middle education, respectively). There were no associations between education and unprocessed red meat consumption in Mexico.

There was no association between income and probability of red meat consumption in Canada. In Mexico, those with low wealth were 13.3 (3.1–23.6) percentage points less likely than those with high wealth to consume unprocessed red meat. In the US, the predicted probability of consuming unprocessed red meat was 4.4 (0.6–8.2) percentage points higher for those with middle income, relative to those with high income. Adjusted mean intake of unprocessed red meat among those who consumed unprocessed red meat did not differ by either wealth and income or education within any country (Appendix A).

### 3.4. Processed Meat

In both Canada and the US, men were more likely to consume processed meat than women (13.4 (10.1–16.7) and 7.7 (4.7–10.6) percentage points higher, respectively) (Appendix A). Among consumers, estimated average processed meat intake was higher for men than women in Canada (71.1 (65.6–76.7) and 51.8 (47.4–56.1) g, respectively) and the US (70.7 (66.2–75.3) and 49.6 (47.2–52.0) g, respectively) but not for Mexico (60.5 (48.5–72.6) and 57.2 (35.8–78.6) g, respectively).

In Canada, those with low and middle education were more likely to consume processed meat than those with high education (4.4 (0.1–8.7) and 6.3 (1.8–10.7) percentage points higher for low and for middle education, respectively). In the US, the predicted probability of processed meat consumption was 6.6 (2.6–10.7) percentage points higher for those with middle education, relative to those with high education. There were no associations between education and likelihood of processed meat consumption in Mexico.

In Canada, individuals with low income were 5.1 (0.8–9.4) percentage points less likely than those with high wealth to consume processed meat. In Mexico, those with low and middle wealth had a lower predicted probability of consuming unprocessed red meat, relative to those with high wealth (14.8 (5.3–24.2) and 11.5 (3.1–19.9) percentage points lower for low and middle wealth, respectively). There were no associations between income and likelihood of processed meat consumption in the US. Adjusted mean intake of processed meat among those who consumed processed meat did not differ by either age, wealth and income or education within any country (Appendix A).

## 4. Discussion

On any given day, between 63 and 74% of individuals consume red or processed meat in North America. Recent public health guidelines, including national dietary guidelines in North America, recommend limiting red and processed meat consumption [5,6,24]. Canada’s 2019 Food Guide encourages plant-based proteins over meat whenever possible, and lists processed meat as an unhealthy food that should not be consumed regularly [5]. Mexico’s dietary guidelines similarly recommend limiting consumption of unprocessed red meat and choosing legumes or white meat when possible [24]. The current Dietary Guidelines for Americans (DGA) treat protein-source foods equally and do not mention limiting red or processed meat as long as the overall diet is healthy with regards to saturated fat and sodium [25].

The recent EAT–Lancet commission went beyond these general recommendations and considered 14 g/day of unprocessed red meat and 0 g/day of processed meat as optimal levels to promote health and reduce disease risk and mitigate climate change. Although it does not mention frequency explicitly, the recommended levels are so low that effectively, they translate to not eating meat daily. The commission’s benchmarks are particularly salient for North America, where alternative high-quality protein sources are readily available such as poultry, low-fat dairy, beans and legumes, and nuts and seeds [6]. Despite these recommendations, in addition to the high proportion of meat consumers, mean levels of unprocessed red meat and processed meat in North America are substantially higher than these benchmarks.

Men were more likely than women to eat red and processed meat in all three countries. Moreover, among those who consumed meat in Canada and the US, mean intake of meat was significantly higher for men than women. Meat is historically associated with maleness across a variety of cultural contexts, and meat consumption is a way to affirm masculinity [26]. The results of this study are consistent with this sociocultural phenomenon and with evidence from other dietary studies throughout North America, which have consistently found that men consume higher levels of red and processed meat [6,13,14,27].

Additionally, although the predicted probability of meat consumption was similar across age groups in Canada and the US, older individuals ate smaller amounts of total meat in these countries. This is consistent with recent studies on meat intake throughout the life course [13,27] and is likely due to smaller overall dietary intake among the oldest populations. Previous studies in Mexico have looked at meat as a contributor to energy intake in the total adult population over 20 years, but there is little information on diet in the elderly [28]. In the results for Mexico, the predicted probability of consuming total meat was much lower for the oldest individuals, 75 and older, which may reflect a cohort effect of a more traditional diet that was common when these individuals were young [7].

In Canada and the US, education was a strong predictor of red and processed meat consumption. In high-income settings that are in the final stages of the nutrition transition, education is a more consistent predictor of dietary quality than measures of wealth or income [29]. Education is closely correlated with health and nutritional literacy, which enables individuals to integrate health and nutrition knowledge into their everyday lives and practices [30]. Dietary improvements in high-income settings have occurred disproportionately among highly-educated and motivated individuals [31]. A similar pattern was observed for red and processed meat in Canada and the US. Those with lower educational attainment were more likely to consume red and processed meat than individuals with higher levels of education. The educational differences in meat consumption may perpetuate diet and health-related disparities in these countries [32] and highlight the need for equitable prevention policies.

For Mexico, on the other hand, wealth was more strongly associated with meat intake. Individuals with lower and middle wealth were less likely to eat meat than the wealthiest individuals. Mexico is an upper middle-income country that is in an earlier phase of the nutrition transition than either Canada or the US [33]. As the economies of middle-income countries such as Mexico expand, people begin to spend more per food item and meat consumption rises, starting with wealthier individuals but eventually extending to the less wealthy [6,31]. Our results were consistent with other findings that in Mexico, those with high socioeconomic status consume more meat products than those with medium or low socioeconomic status [14]. This trend raises concerns for the already-high burden of noncommunicable diseases in these countries [31], as well as for the climate implications of the growing global demand for meat [4,6].

Globalization and market integration have been central in transforming food systems and diets in ways that encourage red and processed meat consumption, particularly for middle-income countries such as Mexico [7]. The economies of Canada, Mexico, and the US are tightly linked via trade agreements. NAFTA eliminated nearly all trade barriers and tariffs for agricultural products, including red and processed meat. Since NAFTA went into effect, trade of red meat products between the three nations has increased substantially [34]. Domestic agricultural subsidies have further encouraged red meat production and kept prices relatively low, particularly for US exports [7].

Together, these policies play a critical role in shaping the patterns of red and processed meat consumption that this study observed across North America. They facilitate high levels of meat intake in high-income settings such as Canada and the US [6,35]. Middle-income countries increasingly account for a greater share of global meat consumption [31], and NAFTA has accelerated this trend in Mexico [7]. Under the trade agreement, Mexican importation of meat—particularly pork—has considerably outpaced its exports. Accordingly, in Mexico per-capita consumption of beef has risen appreciably, and per-capita consumption of pork has more than doubled since 1994 [34]. The ratification of USMCA in March 2020 continues trade protections for meat products and was widely praised by the meat industry [8].

The present study has several strengths. It is the first to compare dietary intake of red and processed meat across North America. The included surveys contain the most recently available nationally-representative dietary intake data for all three countries. Nonetheless, this study has several limitations. Our results are based on a single 24-h dietary recall, which is subject to recall bias and may not represent participants’ long-term dietary intake. However, a single 24-h dietary recall is generally considered acceptable for estimating population mean intake [36]. This is a cross-sectional study, so we cannot comment on trends in meat intake over time. Future research could seek to analyze trends in meat intake over time and its associations with health consequences at a nationally-representative scale. Categories of wealth and income and education were not the same for Canada, Mexico, and the US. However, the country-specific contexts were carefully considered in order to best capture socioeconomic differences within a country. Additionally, the range of evaluated sociodemographic predictors was limited, as we did not have access to an indicator of urbanicity, region, or state, or other characteristics that may be predictive of meat intake, such as political orientation. Furthermore, such characteristics do not have meanings that are comparable across these three countries. Finally, the objective was to evaluate patterns of red and processed meat consumption in North America, but considering the unique cultural contexts and economic relationships between Canada, Mexico, and the US, the results may not be generalizable to other settings. However, given the nationally-representative survey data used, this study offers key insights to population-level correlates of red and processed meat consumption in three countries with large populations, high levels of meat intake, and close economic ties.

## 5. Conclusions

This is the first study to compare individual-level predictors of red and processed meat intake across North America. Consumption of red and processed meat is high in all three countries. Policy solutions are urgently needed to promote public health and improve the sustainability of diets. Taxes are effective tools to reduce intake of sugar-sweetened beverages and junk foods, and could also be used to target intake of red and processed meats [37]. Efforts to increase consumer awareness about the negative health and environmental impacts of meat, such as warning labels, also have the potential to shift behavior [38]. Redirection of meat subsidies towards more sustainable and nutritious crops, such as fruits, vegetables, and legumes, could shift incentives to favor healthier alternatives [38]. National dietary guidelines can also play a role. Canada’s Dietary Guidelines encourage individuals to consider the health and environmental impact of their food choices [5]. Both the Canadian and Mexican dietary guidelines recommend consuming plant-based sources of proteins wherever possible [5,24]. The US DGA are currently being updated, but the US federal government has, for the first time, restricted the science that can be considered in the guidelines, specifically prohibiting consideration of environmental impacts [39]. Aligning national dietary guidelines and trade policies with scientific evidence could bolster—rather than hinder—public health efforts across the continent.

## Figures and Tables

**Figure 1 ijerph-18-00357-f001:**
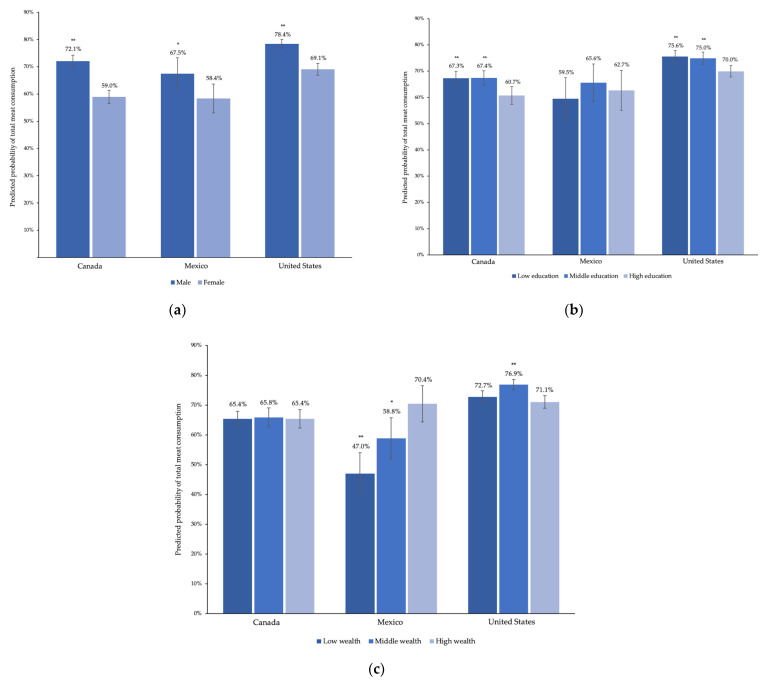
Adjusted predicted probability of total meat consumption by (**a**) sex, (**b**) education, and (**c**) wealth and income for one day of 24-h dietary recall data in Canadian Community Health Survey (2015), Mexico National Health and Nutrition Survey (2016), and US National Health and Nutrition Examination Survey (2013–2016). Multivariate negative binomial models were adjusted for sex, 10-year age category, educational attainment, and wealth and income category; * *p* < 0.01, ** *p* < 0.001 for the observed difference in predicted probability of any meat consumption.

**Figure 2 ijerph-18-00357-f002:**
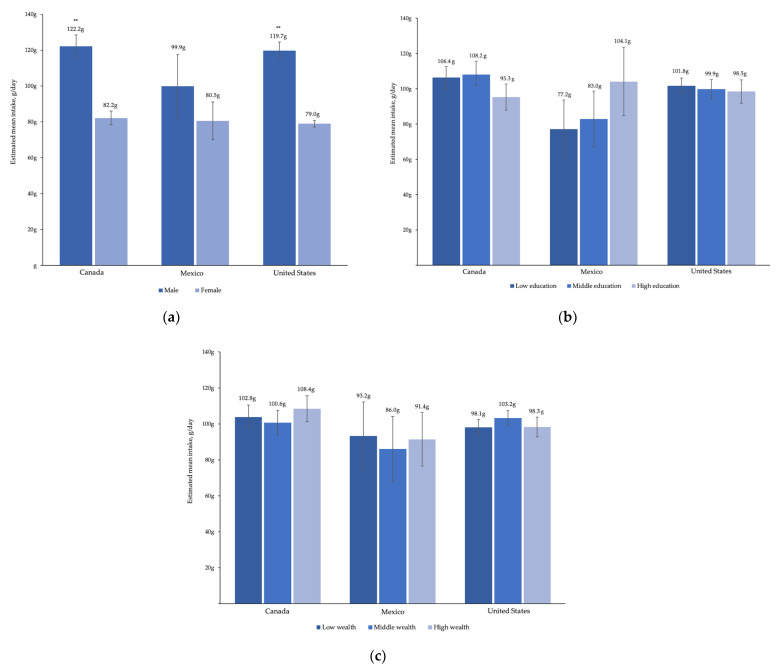
Adjusted estimated mean grams of total meat intake by (**a**) sex, (**b**) education, and (**c**) wealth and income category for one day of 24-h dietary recall data in Canadian Community Health Survey (2015), Mexico National Health and Nutrition Survey (2016), and US National Health and Nutrition Examination Survey (2013–2016). Multivariate generalized linear regression models were adjusted for sex, 10-year age category, educational attainment, and wealth and income category; ** *p* < 0.001 for the observed difference in estimated mean intake of total meat.

**Table 1 ijerph-18-00357-t001:** Characteristics of non-pregnant adults aged ≥18 years with at least one day of 24-h dietary recall data in Canada, Mexico, and the United States (*n* = 26,161).

Characteristics	Canada-CCHS *n* = 14,083	Mexico-ENSANUT *n* = 1581	US-NHANES *n* = 10,497
**Sex ***			
Male	49.7 (6630)	48.1 (608)	49.0 (5101)
Female	50.3 (7453)	51.9 (973)	51.0 (5396)
**Age ***			
18–24	8.5 (1101)	17.0 (354)	12.2 (1363)
25–34	15.7 (1973)	20.3 (250)	17.4 (1685)
35–44	18.8 (2180)	21.0 (313)	16.1 (1704)
45–54	19.4 (2534)	17.8 (234)	18.0 (1704)
55–64	16.3 (2203)	11.3 (208)	16.9 (1725)
65–74	13.1 (2205)	8.3 (137)	11.8 (1341)
75+	8.2 (1887)	4.2 (85)	7.6 (975)
**Educational attainment *^,†^**			
Low	38.6 (6150)	32.9 (674)	36.4 (4541)
Medium	33.5 (4559)	29.1 (461)	33.1 (3114)
High	27.9 (3281)	38.1 (446)	30.5 (2530)
**Wealth and Income *^,‡^**			
Low	45.2 (5173)	19.8 (522)	21.9 (3220)
Medium	28.2 (4560)	27.2 (534)	38.5 (4042)
High	26.7 (4343)	53.0 (525)	39.6 (3235)
**Proportion consumers ^§^**			
Unprocessed red meat	46.1 (44.3–47.9)	45.9 (40.9–51.0)	48.5 (47.1–49.9)
Processed meat	36.3 (34.6–38.0)	30.6 (25.9–35.8)	47.1 (45.7–48.5)
Total meat	65.6 (63.9–67.2)	62.7 (58.1–67.1)	73.6 (72.3–74.8)
**Median (IQR) intake (grams) ^#^**			
Unprocessed red meat	79.0 (36.6–131.6)	62.0 (28.7–114.8)	72.3 (38.3–124.5)
Processed meat	41.8 (21.2–82.4)	40.0 (20.0–76.9)	44.5 (17.9–84.2)
Total meat	79.0 (36.2–140.1)	62.5 (31.3–117.4)	79.4 (40.8–134.7)

* Values are weighted % (unweighted N). Weighted % accounts for complex survey weights; ^†^ Educational attainment was defined as low (high school equivalent or lower), middle (trade or college certificate), or high (university degree or higher) in Canada; low (primary school or lower), middle (greater than primary but less than high school), or high (high school or higher) in Mexico; low (high school equivalent or lower), middle (some college), or high (college degree or higher) in the US. ^‡^ Wealth and income was defined as tertiles of self-reported per-capita household income in Canada. In Mexico, wealth and income was derived from a standard household asset index and divided into low, middle, and high. In the US, wealth and income was derived from the federal poverty to income ratio (PIR): low (PIR 0–100%), middle (PIR 101–399%), and high (PIR ≥ 400%). ^§^ Values are weighted % (95% CI) and account for complex survey weights. ^#^ Values are weight in grams consumed among consumers and account for complex survey weights.

## Data Availability

The datasets analysed during the current study are publically available for download at: https://www150.statcan.gc.ca/n1/en/catalogue/82M0024X2018001 (Canada, CCHS), https://ensanut.insp.mx/encuestas/ensanut2016/descargas.php (Mexico, ENSANUT), https://wwwn.cdc.gov/nchs/nhanes/Search/DataPage.aspx?Component=Dietary&CycleBeginYear=2013, https://wwwn.cdc.gov/nchs/nhanes/Search/DataPage.aspx?Component=Dietary&CycleBeginYear=2015 (United States, NHANES).

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
