# Peer review of "Patterns of Red and Processed Meat Consumption across North America: A Nationally Representative Cross-Sectional Comparison of Dietary Recalls from Canada, Mexico, and the United States"

_ijerph, 2021, doi:10.3390/ijerph18010357_

Round 1

Reviewer 1 Report

STRUCTURE

The manuscript is properly structured.

TITLE

The title should inform that the type of study.

ABSTRACT

The abstract should inform that the type of study.

INTRODUCTION

  • Line 40: Review the reference and correct [
  • Line 48: what is the global average?
  • Do not use the first person in scientific publications. Applicable to the entire document (lines 54, 61, 107, 110, …)
  • State specific objectives.
  • No hypotheses are included.

MATERIAL AND METHODS

  • Indicate the type of study.
  • Present key elements of study design early in the paper
  • Clearly define all outcomes, exposures, predictors, potential confounders and effect modifiers
  • Describe comparability of assessments methods
  • Describe any efforts to address potential sources of bias.
  • Line 111-112: The correct way to abbreviate gram/day is g/day
  • Do not use the first person in scientific publications

RESULTS

  • Consider use of a flow diagram

DISCUSSION

  • Summarise key results with reference to study objectives
  • Line 264: Choose g/day or gram/day, not both
  • Line 265: Add reference
  • Discuss the generalisability (external validity) of the study results
  • Do not use the first person in scientific publications

CONCLUSIONS

  • Do not use the first person in scientific publications

REFERENCES

  • Not all references follow the style indicated.

Reviewer 2 Report

This study estimates the patterns of red and processed meat consumption in three north American countries. The paper is easy to understand, but more analysis between the meat consumption and health issue is needed. My specific comments are as follows:

First, the current study only estimates the probability and mean of red meat for various subgroups. I think further analysis is needed to link the consumption of red meat with health issues. For instance, what’s the association between consumption of red meat and obesity?

Second, usually we use household per capita income rather than total income to define income tertiles, as the total income highly depends on household size and structure.

Third, the definition of education attainment quintile in the US is missing in supplementary Tables.

Fourth, since you already report results from fully-adjusted models, results from univariate analyses can be removed.

Reviewer 3 Report

This is a cross-sectional study discussing about whether there was risk factor related to patterns of red and processed meat consumption across North America. This study used a population-based, nationally-representative 24-hour dietary recall data from Canada, Mexico, and the US. I think the topic is important and contributive to the public health with an empirical approach quite valuable for public policy professionals.

Major concerns:

1. The research appropriately used negative-binomial model to analyze 24-recall data. Based on my knowledge, dispersion test should be provided to approval evidence of selected method in this study. By the way, exponential model also had the similar results?

2. Table 1 provided some characters of studied subjects. The authors should provide more general/additional characters of studied subjects.

3. Please use power analysis to statement adequate sample size in this study.

4. Lastly, the authors only briefly discuss limitations, acknowledging that the main limitation is the cross-sectional design. They should elaborate on how the use of this design is subject to Incidence-Prevalence bias, also known as Neyman bias, and how that might influence their findings.

5. Some references are outdated and should be updated accordingly.

Round 2

Reviewer 1 Report

No further comments. 

Author Response

Thank you for your reviews. We value your input!

Reviewer 2 Report

Authors have well addressed all my comments. I have no further comments.

Author Response

(The authors gave the same response as above.)

Reviewer 3 Report

Thanks for great efforts on revision. Please improve resolution of figures in main text while figures enlarged. Move figure s1 and add corresponding description to the main text to provide readability to researchers if you agree.

Author Response

Please improve resolution of figures in main text while figures enlarged.

Response: Thank you for your comment. All figures Additionally, all figures have been provided to IJERPH in a zip archive so that the journal's editorial team may incorporate them appropriately into the final version.

Move figure s1 and add corresponding description to the main text to provide readability to researchers if you agree.

Response: Thank you for your comment. We now reference Supplementary Figure S1 in line 164.

Thank you kindly for your reviews. We value your input!